# The Charophytes (Charophyceae, Characeae) from Dagestan Aquatic Habitats, North Caucasus: Biogeographical and Barcoding Perspectives

Roman E. Romanov [1], Maxim M. Mallaliev [2], Sophia Barinova [3,*], Vyacheslav Yu. Nikulin [4] and Andrey A. Gontcharov [4]

1 Komarov Botanical Institute of the Russian Academy of Sciences, Professora Popova Str., 2, 197376 St. Petersburg, Russia; romanov_r_e@ngs.ru
2 Mountain Botanical Garden of the Dagestan Federal Research Centre of the Russian Academy of Sciences (MBG DFRC RAS), Gadgiev St., 45, 367000 Makhachkala, Russia; max-im.mallaliev@yandex.ru
3 Institute of Evolution, University of Haifa, Abba Khoushi Ave, 199, Mount Carmel, Haifa 3498838, Israel
4 Federal Scientific Center of the East Asia Terrestrial Biodiversity, Far Eastern Branch of the Russian Academy of Sciences, 100-letiya Vladivostoka Ave, 159, 690022 Vladivostok, Russia; nikulinvyacheslav@gmail.com (V.Y.N.); gontcharov@biosoil.ru (A.A.G.)
* Correspondence: sophia@evo.haifa.ac.il

**Abstract:** The charophytes in many regions of the world are still poorly understood. This hampers the synthesis of distributional and ecological datasets at worldwide and continental scales, as well as complicates the generalization of species concepts for widely distributed and local taxa. To fill in the blanks for charophytes in the Caucasus and to improve our knowledge of species distribution areas in Eurasia, a field survey and study of available specimens from Dagestan (North Caucasus, Russia) was conducted based on morphological observation using light and scanning electron microscopy and molecular genetic analyses allowing for the precise identification and testing of the presence of cryptic and undescribed taxa. Nineteen new localities for seven *Chara* species and one *Tolypella* species, seven new species, and one new genus were identified in the studied region, and one new species in the Caspian Sea region was found. Some species records changed the outline or filled in the gaps in species distribution data. The presence of species distributed mainly in central Eurasia (*C. globata*, *C. neglecta*) with mainly Mediterranean–Middle Eastern species (*C. gymnophylla*) is notable for this region, as well as for other studied regions of the Caucasus characterized by a mixture combination of species with different distribution patterns. *Chara gymnophylla* was frequent in Dagestan, similar to the Mediterranean and Middle Eastern regions. Small brackish waterbodies on the coast of the Caspian Sea, freshwater mountain rivers, small associated waterbodies, and water reservoirs are the main habitats of charophytes in the studied region. Based on habitat preference and distribution in the Caucasus, recommendations for the protection of some species were suggested. The lack of endemic species among charophytes from Dagestan and Caucasus contrasts with the flora of terrestrial magnoliophytes that is rich in species endemism.

**Keywords:** characeae; *Chara*; *Tolypella*; Dagestan; Caucasus; Caspian sea; DNA barcoding; protection

## 1. Introduction

The charophytes comprise a distinct group of macroscopic algae widely known mainly as ecosystem engineers and pioneer species [1]. Their life strategies and ecology are diverse [2–4] and still poorly studied for many species, especially outside Europe and Australia. Regional surveys of charophytes are an essential step toward better understanding their distribution and ecology. They allow further testing and improvement of species concepts and clarification of the bioindication potential of charophytes. The DNA barcoding of widely and narrowly distributed species fits the same targets, allowing testing of the

presence of cryptic and undescribed taxa. Lesser known and remote areas are especially important from this perspective.

Some species of charophytes are typical for the protected freshwater habitat types "Hard oligo-mesotrophic waters with benthic stoneworts *Chara* spp." of the Habitat Directive from the European Union and "C1.2a. Permanent oligotrophic to mesotrophic waterbody with Characeae", a vulnerable one according to the European Red List of Habitats [5]. The charophytes are one of the most threatened groups of plants in Europe [6], easily illustrated by their proportionally high representation among endangered species on European and national Red Lists [7,8]. The key areas important for the protection of vulnerable and endangered species were outlined in only a few European countries [9,10].

The current approach in taxonomical studies of charophytes is detailed morphological studies using light and scanning electron microscopy [11–20], usually in combination with phylogenetic analyses [21–41]. Different regions of Eurasia were the focus of floristic research on charophytes in the last decade [10,42–76], covering distribution, ecology, and protection, but only Europe has contemporary comprehensive flora mostly based on integrative taxonomy for its whole territory [77].

Knowledge about the charophytes of Dagestan, a region east of the North Caucasus, is limited and almost unknown worldwide. All available data are basically limited to three old specimens of *Chara vulgaris* L. from rivers Akusha and Yaloma and small water bodies near the village of Kajagent collected by Th. Alexeenko and reported in two articles covering large areas of Northern and Central Eurasia [78,79], later summarised by Hollerbach [80]. A few recent records, mainly at the genus level (*Chara* L.), are available at iNaturalist.org [81]. A similar situation could be noted for the entire North Caucasus, based on a longer list of references with only a few records in each of them (see note for Table 1 below).

As a first step towards better understanding this group of freshwater macrophytes, a keystone in some ecosystems and notable for their ecological role, we present here the main results of some recent field and herbarium investigations. In previous studies of Caucasian charophytes, species identification was carried out using only a morphological approach, which did not allow precise species identification in all cases. The aim of this study was to investigate the species composition of the charophytes of Dagestan using a polyphasic approach, which included morphological observations and molecular genetic analyses of the DNA of the studied species according to contemporary charophyte research standards.

## 2. Materials and Methods

### 2.1. Morphological Identification

The specimens were usually collected by hand during a careful survey of water bodies. They were dried as herbarium specimens and stored in LE (acronyms according to [82]). Some specimens collected in the 19th and 20th centuries were found in LE. Almost all of them had no identification before this study. Our efforts to search charophyte specimens from Dagestan stored elsewhere yielded no results. More than 30 pressed specimens were examined in the present study. A few records of charophytes were available at iNaturalist.org [81], but only two can be identified at the species level (*Chara globata* Migula).

The morphological features of the specimens were studied using an Olympus SZ61 stereomicroscope (Olympus Corporation, Shinjuku, Tokyo, Japan). Photographs of diagnostic traits were taken using a digital camera. Oospores taken from some recent specimens for scanning electron microscopy (SEM) were treated according to a previously described method [26]. The cleaned oospores were coated with gold and studied using a Jeol JSM 6390LA scanning electron microscope (JEOL Ltd., Tokyo, Japan). Taxonomy followed the most recent reference [77]. The Ecoregions' mapping program was used to map the individual charophyte species distribution in the studied territory [83]. The BioDiversity Pro 2.0 program was used for the similarity calculation [84].

### 2.2. DNA Extraction, Amplification, and Sequencing

Total genomic DNA was extracted as described previously by Echt et al. [85] with some modifications [86]. Part of the *rbc*L gene was amplified as described previously [37]. The PCR products were purified with ExoSAP-IT PCR Product Cleanup Reagent (Affymetrix Inc., Santa Clara, CA, USA) and sequenced in both directions using an ABI 3500 genetic analyzer (Applied Biosystems, Waltham, MA, USA) with a BigDye terminator v. 3.1 sequencing kit (Applied Biosystems, Waltham, MA, USA). Sequences were assembled using the Staden Package v.1.4 [87] and aligned manually in the SeaView program [88]. The *rbc*L sequences were deposited in GenBank (*C. connivens* OQ607406; *C. neglecta* OQ607411; *C. neglecta* OQ607412; *C. globata* OQ607413; *C. gymnophylla* OQ607410; *C. gymnophylla* OQ607409; *C. gymnophylla* OQ607408; *C. vulgaris* var. *longibracteata* OQ607407; Supplementary Materials).

Before the phylogenetic analyses, the sequences of the *rbc*L gene were compared with those available at the National Center for Biotechnology Information (NCBI, Bethesda, MD, USA) using a BLAST search [89] to estimate their taxonomic position.

### 2.3. Phylogenetic Analyses

The *rbc*L dataset was used to access the affinity of our *Chara* species with the genus representatives retrieved from the NCBI. This dataset was assembled as described by Romanov et al. [37,38]. Maximum likelihood (ML) analyses were carried out using PAUP 4.0b10 [90]. Bayesian inference (BI) was performed using MrBayes 3.1.2 [91]. To determine the most appropriate DNA substitution models for our datasets, we used the Akaike information criterion (AIC; [92]), which was applied using jModelTest 2.1.1 [93]. The GTR+I+G model was selected as the best fit for the *rbc*L dataset. ML analyses were carried out using heuristic searches with a branch-swapping algorithm (tree bisection-reconnection). Using BI, four parallel MCMC runs were carried out for 3 million generations. Sampling was carried out every 100 generations for a total of 30,000 samples. The convergence of these two chains was assessed, and stationarity was determined according to the 'sump' plot (the first 25% of samples were discarded as 'burn-in'). The posterior probabilities were calculated from the trees sampled during the stationary phase. The robustness of the trees was estimated based on bootstrap percentages (BP; [94]) in ML and posterior probabilities (PP) in BI. A BP < 50% and PP < 0.95 were not considered. An ML-based bootstrap analysis was inferred using the web service RAxML version 7.7.1 (http://embnet.vitalit.ch/raxml-bb/; [95]; accessed on 1 March 2023).

## 3. Results

### 3.1. Charophyte Diversity and Distribution

Eight species of charophytes, including seven species of *Chara* and one species of *Tolypella* (A.Braun) A.Braun, were found at 24 localities in Dagestan according to all available data (see below). The presence of a few taxa known before from this territory, i.e., *C. vulgaris* var. *vulgaris* and var. *longibracteata* (Kütz.) Kütz. [78,79] was confirmed with the studied specimens. Twenty two localities of charophytes were found for the region, and all of them were based on vouchers available for study.

All localities were situated in two ecoregions: Pontic steppe and Caucasus mixed forests (Figure 1b,c). The first region is rich in both species and localities. All species except *C. contraria* A.Braun ex Kütz. were previously known. *Chara connivens* Salzm. ex A.Braun, *C. contraria*, *C. globata* Migula, *C. globularis* Thuill., *C. gymnophylla*, A.Braun, and *C. vulgaris* were found in a few water bodies in Caucasus mixed forests.

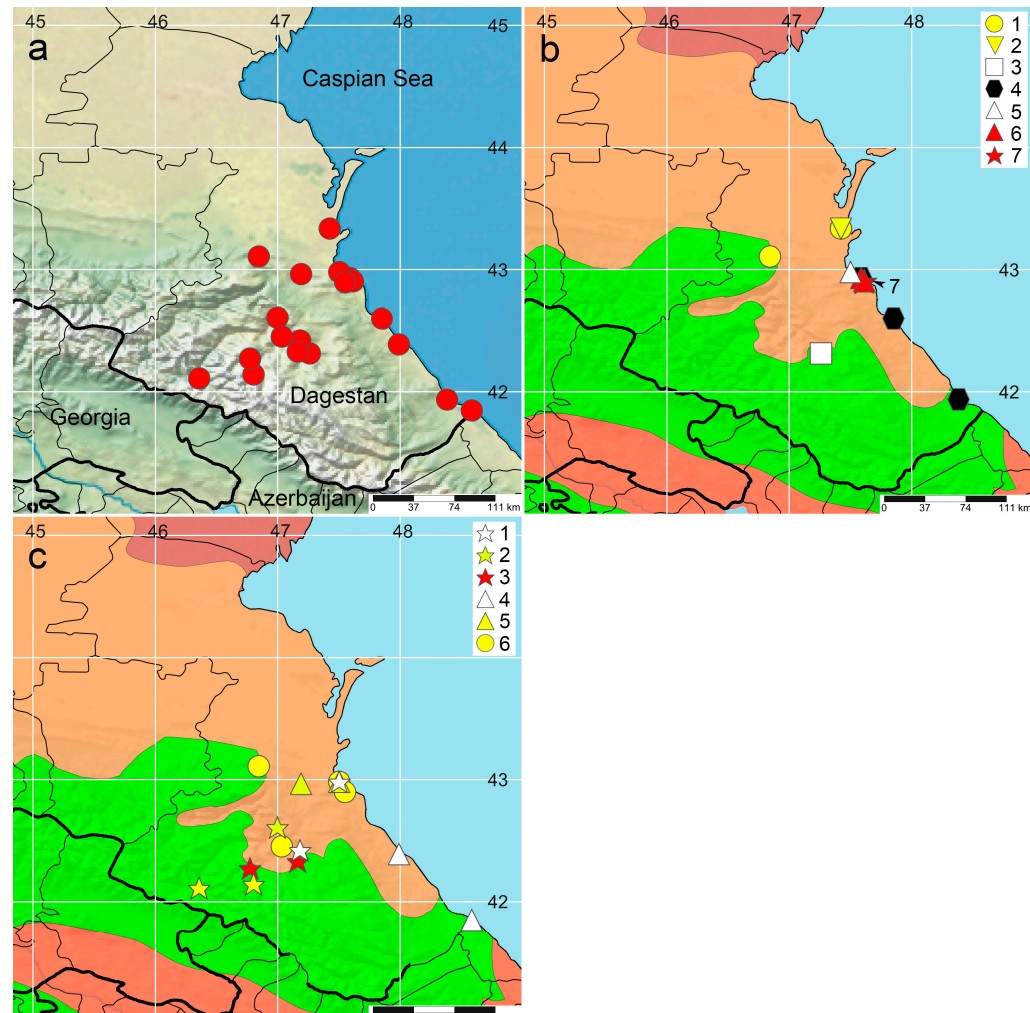

**Figure 1.** Species distribution in the context of elevation (**a**) and ecoregions (**b,c**) in the region studied at three sampling intervals: a–all species, b: 1—*Chara connivens*, 20th century, 2—*C. globularis*, 20th century, 3—*C. contraria*, 19th century, 4—*C. globata*, 21st century, 5—*C. neglecta*, 19th century, 6—*C. neglecta*, 21st century, 7—*Tolypella nidifica*, 21st century, c: 1—*C. gymnophylla*, 19th century, 2—*C. gymnophylla*, 20th century, 3—*C. gymnophylla*, 21st century, 4—*C. vulgaris* var. *vulgaris*, 19th century, 5—*C. vulgaris* var. *vulgaris*, 20th century, 6—*C. vulgaris* var. *longibracteata*, 20th century. Ecoregions [83]: dark brown–Caspian lowland desert, light brown–Pontic steppe, green–Caucasus mixed forests.

Distribution data, the habitat, and floristic novelty for the species found in the studied area are listed below. *Chara connivens*, *C. contraria*, *C. globata*, *C. globularis*, *C. gymnophylla*, *C. neglecta* Hollerb., and *Tolypella nidifica* (O.F.Müll.) A.Braun are new species records for Dagestan. *Chara neglecta* is the second species record reported for the Caucasus. *Tolypella nidifica* is the first species and genus record for the Caspian Sea region.

*Chara connivens* Salzm. ex A.Braun (Figures 1b and 2a)

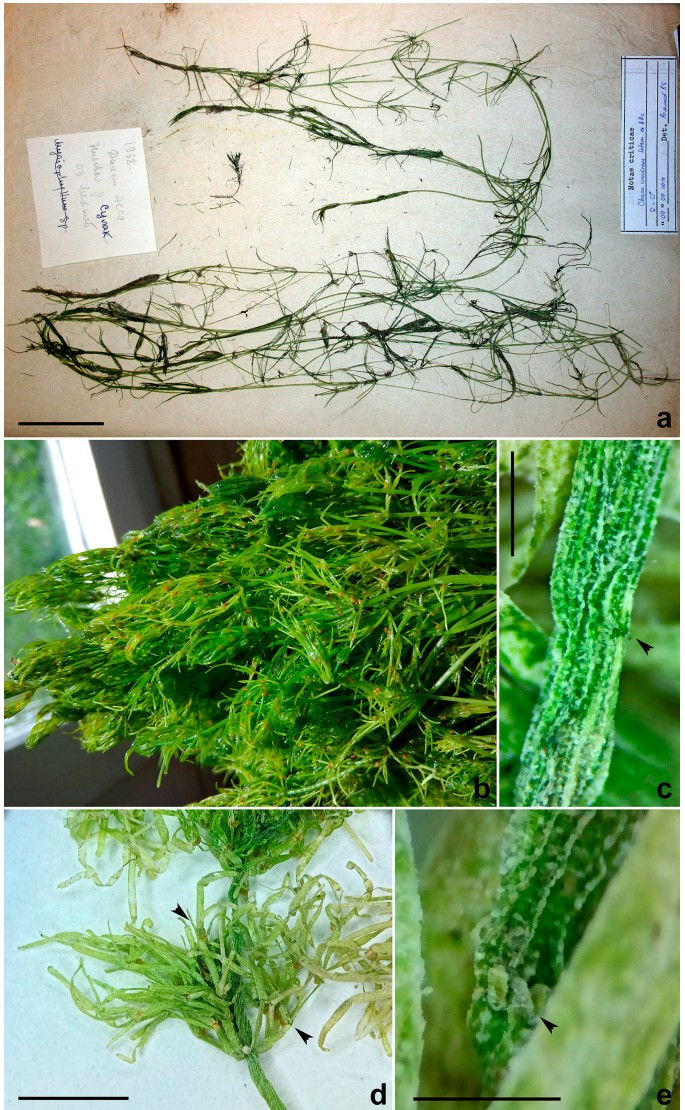

**Figure 2.** *Chara connivens* and *C. gymnophylla* from Dagestan (LE): (**a**)—the general habit of pressed *C. connivens* from the Mekhteb Reservoir showing elongated general appearance and long slender branchlets; (**b**–**e**)—*C. gymnophylla* from the stagnant water body on the bank of the Karalazurger River: (**b**)—general appearance of living plant; (**c**,**e**)—diplostichous aulacanthous stem cortex with tiny ((**c**), arrowhead) and slightly elongate solitary spine cells ((**e**), arrowhead); (**d**)—whorl of well-developed completely ecorticate branchlets with long adaxial bract cells and conjoined gametangia (arrowheads). Scale: (**a**)—5 cm, (**c**,**e**)—1 mm, (**d**)—4 mm. Photos: (**a**,**c**–**e**)—by R.E. Romanov, (**b**)—by M.M. Mallaliev.

Studied specimens: 1. [Gergebilsky District], River Sulak, Chiryurt Reservoir, at shallow, in a pit, 8 August 1968, V.M. Katanskaya (LE A0002061). 2. [Babayurtovsky District] lower reach of the River Sulak, Lake Mekhteb [currently Mekhteb Reservoir], together with *C. globularis* Thuill., 1968, V.M. Katanskaya (LE).

Habitat: large mountain and coastal water reservoirs.

*Chara contraria* A.Braun ex Kütz (Figure 1b).

Studied specimen: *Dagestan, distr. Dargi. In paludosis ad fl. Akuscha inter pagos Urkhuwah et Urchaczi* [currently River Akusha or Akushinka, vicinity of the rural locality of Urkhuchi-makhi (Urkhuchi)], 4000′ [1219 m a.s.l.], [together with *C. vulgaris* in Ch. 64], 16 July 1898, Th. Alexeenko. *Flora Caucasi / C. foetida* Al. Br., [det. J. Vilhelm] (LE Ch 63–65). This locality was cited for *C. vulgaris* [76].

Habitat: wetlands associated with mountain rivers.

*Chara globata* Migula (Figures 1b and 3a–f)

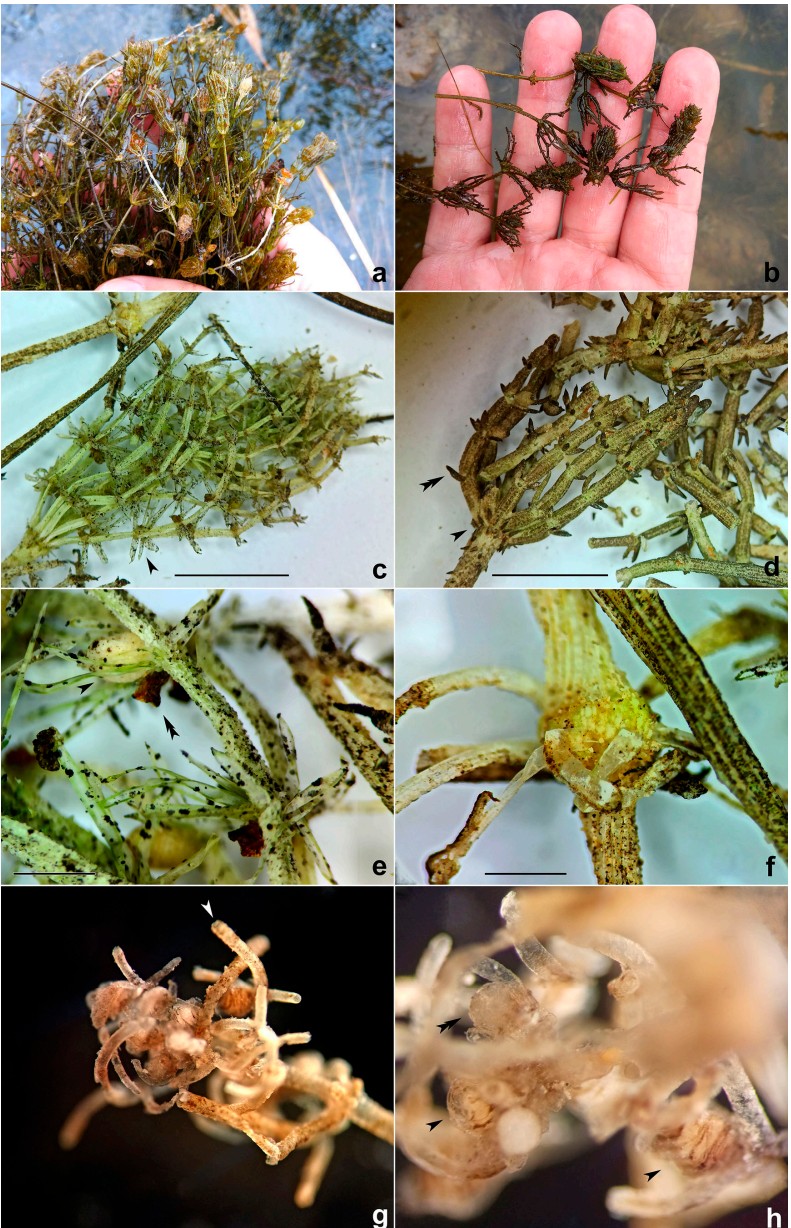

**Figure 3.** *Chara globata* and *Tolypella nidifica* from water bodies on the coast of the Caspian Sea (LE): (**a**,**c**)—upper parts of first morphotype of *C. globata* lacking evident lime incrustation and having long abaxial bract cells (arrowhead); (**b**)—upper part of the second morphotype of *C. globata* showing evident lime incrustation and abaxial bract cells of moderate length; it is really similar to *C. baltica* (Hartm.) Bruzelius and differs with strong lime incrustation only; (**c**)—whorl of branchlets of second morphotype of *C. globata* showing acute stipulodes of moderate length (arrowhead), abaxial bract cells of moderate length (double arrowhead) and mainly solitary aculeate spine cells of moderate length; €—nodes of branchlet of first morphotype of *globata* showing corticated segments, long verticillate bract cells and solitary conjoined gametangia (oogonium–arrowhead, antheridium– double arrowhead); (**f**)—starchless nodal bulbils of first morphotype of *globata*; (**g**)—apical part of rewetted specimen of *T. nidifica* showing obtuse end cells of branchlets (arrowhead); (**h**)—fertile head of rewetted specimen of *T. nidifica* showing presence of oogonia (arrowheads) and at least one antheridium (double arrowhead). Both morphotypes of *globata* were found in the same lake near the settlement of Turali. Scale: (**c**,**d**)—5 mm, (**e**,**f**)—1 mm. Photos: (**a**,**b**)—by M.M. Mallaliev, (**c**–**h**)—by R.E. Romanov.

Studied specimen: Coast of the Caspian Sea, Makhachkala, the settlement of Turali, small lake, 42.9317 N, 47.5865 E, 25 m b.s.l., together with *C. neglecta*, 28 June 2020, M.M. Mallaliev (LE).

Records at iNaturalist [81]: 1. Derbentskiy District, vicinity of the village of Arablyar, an oxbow of the Rubas River near the mouth at the coast of the Caspian Sea, 41.9381 N, 48.3816 E, 3 May 2023, https://www.inaturalist.org/observations/159590371. 2. Kayak-entskiy District, northern of the town of Izberbash, small coastal water body, abundant, 42.6005 N 47.8506 E, 5 June 2023, https://www.inaturalist.org/observations/166546785 (accessed on 10 June 2023).

Habitat: small coastal brackish water bodies.

*Chara globularis* Thuill. (Figure 1b)

Studied specimen: [Babayurtovsky District] lower reach of the River Sulak, Lake Mekhteb [currently Mekhteb Reservoir], together with *C. connivens*, 1968, V.M. Katanskaya (LE).

*Chara gymnophylla* A.Braun (Figures 1c, 2b–e and 4a–c)

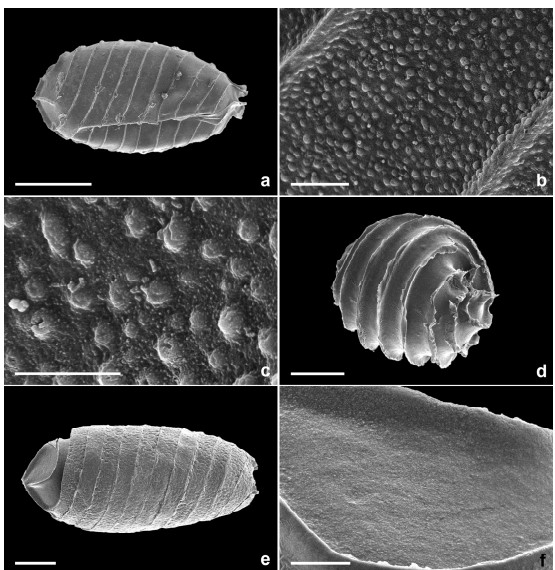

**Figure 4.** Oospores and gyrogonites of *Chara gymnophylla*, *C. neglecta,* and *Tolypella nidifica* (LE), SEM: (**a**–**c**)—oospore of *C. gymnophylla* from the stagnant water body at the bank of the Karalazurger River: (**a**)—later view (ruptured during preparation), showing low ridges and absence of basal cage; (**b**,**c**)—surface of fossae, pustulate and densely minutely granulate between pustulae; (**d**)—tangential view of oospore of *T. nidifica* from the small stagnant water body near the town of Kaspiysk, showing smooth surface and prominent ridges; (**e**,**f**)—gyrogonite and oospore surface of *C. neglecta* from the same water body: (**e**)—lateral view of gyrogonites missing apical part of lime shell, an apex of oospore with low ridges is visible; (**f**)—roughened surface of oospore fossa. Scale: (**a**)—200 μm, (**b**)—10 μm, (**c**)—5 μm, (**d**,**e**)—100 μm, (**f**)—20 μm. Photos by R.E. Romanov.

Studied specimens: 1. Caucasus orientalis. Dagestania borealis, pr. Chodshalmaihi [currently Khadzhalmakhi] ad calarzailem, 500–600 hex [800–960 m a.s.l.], 22 June 1861, Ruprecht (LE A0001459). 2. Dagestan. obl. [Dagestan Oblast], Aul Gunib, 17 May 1890, W. Lipsky. Flora Caucasica (LE A0001311). 3. Dagestan, Gunib Okrug, the valley of the river Karasu–Koysu [currently Karakoisu, called Tleiseruch in its middle reach] between settlements of Yryb and Ylyb, lateral bay of river, ca. 1700 m a.s.l., 31 August 1929, A. Poretzky. Plantae dagestanicae anno 1929 collectae. Geobotanic Expedition of the Institute of Dagestanian Culture of 1929 (LE). 4. Untsukulsky District, near the aul of Arakany [Arakani], road to pass, beyond the gorge, near waterfall, elevation ca. 400 m, 29 August 1953, Ya.I. Prokhanov, N.T. Cheldyshev. Dagestan Agricultural Institute. Ya.I. Prokhanov, Plants of North Dagestan, No. 386a/*Chara gymnophylla* A.Braun f. *subnudifolia* Mig., det. M.M. Hollerbach, 26 February 1955 (LE). 5. Caucasus, Dagestan, Tlyaratinsky District,

the village of Tlyarata, southeastern part, in a gorge, in shallow flowing water, 25 July 1960, Dzhafarov (LE). 6. Levashi District, vicinity of the village of Tsudakhar, Sana River, 42.3291 N, 47.1613 E, 1124 m a.s.l., 8 July 2020, M.M. Mallaliev (LE). 7. Charodinsky District, vicinity of the village of Gochada, stagnant water body at the bank of the Karalazurger River, 42.2716 N, 46.7718 E, 1506 m a.s.l., 29 July 2020, M.M. Mallaliev (LE).

Habitat: mostly water bodies associated with mountain rivers, slowly flowing reaches of mountain rivers.

*Chara neglecta* Hollerbach (Figures 1b, 4e,f and 5)

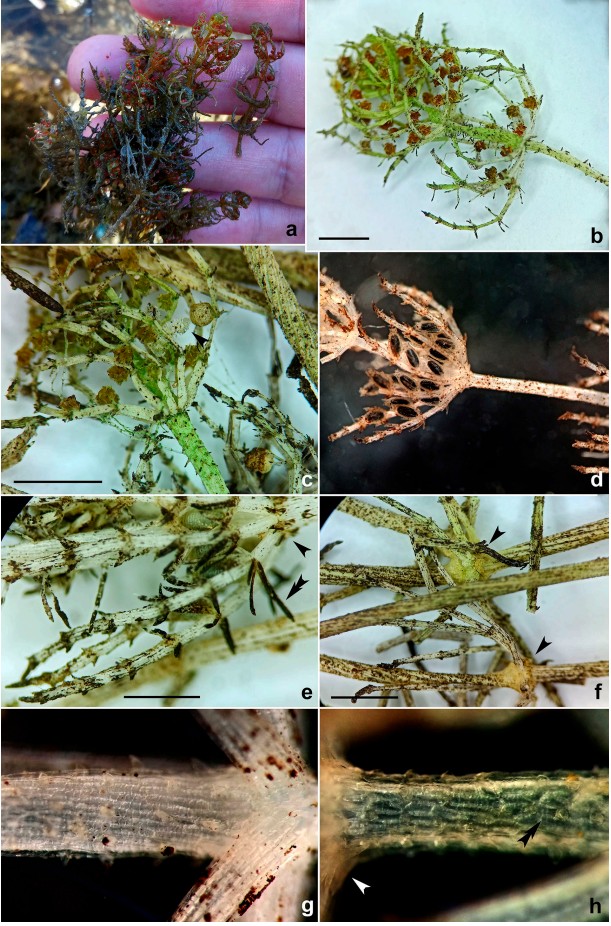

**Figure 5.** *Chara neglecta* from a small water body on the coast of the Caspian Sea near the town of Kaspiysk (LE): (**a**)—upper parts of living robust male plants showing arcuate upper branchlets; (**b**,**c**)—upper part of pressed male plants showing arcuate completely corticated branchlets with solitary antheridia and verticillate bract cells, sparsely short spiny stems, aculeate diplostephanous stipulodes of moderate length (arrowhead indicates triangular shields of antheridium); (**d**)—well-developed whorl of branchlets of female plant (rewetted specimen) having smaller and slender appearance in comparison with male plants (sexual dimorphism), showing completely corticated branchlets with verticillate bract cells of moderate length and solitary oogonia with ripe black oospores, short aculeate stipulodes, short spine cells; (**e**)—branchlet and base of branchlet whorl of pressed female plant showing aculeate diplostephanous stipulodes of moderate length, completely corticated branchlets with short end cells, well-developed verticillate bract cells (double arrowhead), whose length gradually decrease from the base to the tip of branchlet; (**f**)—nodal starchless bulbils (arrowheads) of pressed plants; (**g**)—isostichous triplostichous stem cortex with short solitary acute spine cells (rewetted plant); (**h**)—short aculeate diplostephanous stipulodes (arrowhead) of rewetted plant and irregular diplo-triplostichous stem cortex, slightly aulacanthous (double arrowhead). Scale: (**b**,**c**,**f**)—2 mm, (**e**,**g**)—1 mm. Photos: (**a**)—by M.M. Mallaliev, (**b**–**h**)—by R.E. Romanov.

Studied specimens: 1. Dagestan, Petrovsk [Makhachkala], 2 July 1891, W. Lipsky/ *C. aspera* Willd., det. R. Romanov, 7 October 2015 (LE). 2. Coast of the Caspian Sea, the town of Kaspiysk, in a small stagnant water body, 42.9048 N, 47.6117 E, 21 m b.s.l., together with *Tolypella nidifica* (O.F.Müll.) A.Braun, 7 June 2020, M.M. Mallaliev (LE). 3. Coast of the Caspian Sea, Makhachkala, the settlement of Turali, small lake, 42.9317 N, 47.5865 E, 25 m b.s.l., together with *C. globata*, 28 June 2020, M.M. Mallaliev (LE).

Habitat: small coastal brackish water bodies.

*Chara vulgaris* L. var. *vulgaris* (Figure 1c)

Studied specimens: 1. Dagestan, [no locality], Flora Caucasica No. 4428 (LE). 2. Makhachkala, bog, in a ditch, 25 May 1953, Prokhanov, No. 113 (LE).

Studied specimens: 1. Dagestan, distr. Dargi. In paludosis ad fl. Akuscha inter pagos Urkhuwah et Urchaczi [currently River Akusha or Akushinka, vicinity of the rural locality of Urkhuchimakhi (Urkhuchi)], 4000′ [1219 m a.s.l.], [together with *C. contraria*], 16 Jul 1898, Th. Alexeenko. Flora Caucasi/*C. foetida*, [det. J. Vilhelm] (LE Ch. 64). 2. Prov. Dagestan, distr. Kurink. In littore ad ostium fl. Jaloma [Yaloma River], –80′ [24 m b.s.l.], 11 Aug 1899, Th. Alekseenko. Flora Caucasi No. 3530/*C. foetida* f. *longibracteata* Mig., det. J. Vilhelm, 1929 (LE 71). 3. Prov. Dagestan, distr. Kaitag–Tabassaran. Pr. st. Kajakent [currently Kajagent]. In fossis humidis, –20 [6 m b.s.l.], 17 Jul 1900, Th. Alekseenko. Flora Caucasi No. 4428/*C. foetida* f. *condensata* A.Braun, [det. J. Vilhelm] (LE) (LE). 4. Buynak District, the village of Kapchugay, Shura-Ozen River [Shuraozen River], in water, 2 May 1955, Ya.I. Prokhanov. Dagestan Agricultural Institute. Ya.I. Prokhanov, Plants of North Dagestan, No. 73/det. M.M. Hollerbach (LE). 5.1. Makhachkala District, Makhachkala I, bog, in a ditch near radio station, in water, 4 May 1955, Ya.I. Prokhanov. Dagestan Agricultural Institute. Ya.I. Prokhanov, Plants of North Dagestan, No. 113/det. M.M. Hollerbach (LE). 5.2. The same, in a ditch, 16 Jul 1953, No. 1109 (LE).

Published records: 1. "Caucasus. Dagestan, distr. Dargi. In paludosis ad fl. Arkuscha inter pagos Urkhuwah et Urchaczi 4000′ (10 et 16 July 1898, leg. Th. Alexeenko)" (under the name of *C. foetida* A.Braun; [78]). This gathering contains *C. contraria* and *C. vulgaris* (see above). 2. "Prov. Dagestan, distr. Kaitag–Tabassaran. Pr. st. Kajakent [currently Kajagent]. In fossis humidis (17 July 1900, leg. Alekseenko)" (under the name of *C. foetida* f. *condensata* A.Braun; [79]).

Habitat: small lowland water bodies.

*Chara vulgaris* var. *longibracteata* (Kütz.) Kütz. (Figure 1c)

Studied specimens: 1. Makhachkala District, Makhachkala, bog, in a ditch, 25 May 1953, Ya.I. Prokhanov. Dagestan Agricultural Institute. Ya.I. Prokhanov, Plants of North Dagestan, No. 595 (LE)/*C. vulgaris*, det. M.M. Hollerbach (LE). 2. Makhachkala District, bog, near the railroad station of Tarki [in the settlement of Noviy Khushet], Cherkes-Ozen River, in water, 15 Sep 1953, Ya.I. Prokhanov. Dagestan Agricultural Institute. Ya.I. Prokhanov, Plants of North Dagestan, No. 1301 (LE)/*C. vulgaris*, det. M.M. Hollerbach (LE). 3. [Kizilyurtovsky District] Chiryurt Reservoir at the Sulak River, in the bay on the right coast, at a depth of 0.3 m, at an oozy–sandy bottom, 14 August 1958, V.M. Katanskaya (LE). 4. [Gergebilsky District], Gergebil Reservoir, at shallow, in water from spring, 7 Aug 1968, V.M. Katanskaya (LE A0002061).

Published record: "*Prov. Dagestan, distr. Kurink. In littore ad ostium fl. Jaloma* [Yaloma River] *(11 August 1899, leg. Alekseenko)*" (under the name of *C. foetida* f. *longibracteata* Mig.; [79]). This specimen belongs to var. *vulgaris* (see above).

Habitat: small lowland water bodies, bays, and shallows of mountain water reservoirs, the mouth of lowland rivers.

*Tolypella nidifica* (O.F.Müll.) A.Braun (Figure 1b, Figure 3g,h and Figure 4d)

Studied specimen: Coast of the Caspian Sea, the town of Kaspiysk, in a small stagnant water body, 42.9048 N, 47.6117 E, 21 m b.s.l., together with *C. neglecta*, 7 June 2020, M.M. Mallaliev (LE).

Habitat: small coastal brackish water bodies.

A few other records of *Chara* are known from Dagestan [81], but they cannot be identified at the species level according to available photos.

### 3.2. Species Identification and Phylogenetic Analysis

The identification of some *Chara* samples was based on a combination of morphological and molecular genetic approaches. DNA was isolated from most species, but our efforts were successful for only eight samples (five species: *C. connivens*, *lobateata*, *C. gymnophylla*, *C. neglecta*, *C. vulgaris* var. *longibracteata*). The topologies of the ML and BI trees based on the *rbc*L dataset were similar to that of the BI tree, except for some differences in clade support (Figure 6). Samples of *C. connivens*, *C. vulgaris* var. *longibracteata*, and *lobateata* were placed within species clades (60/0.96, –/– and 65/–, respectively), being part of the species haplotypes (Supplementary Materials). Samples of *C. neglecta* fell in shared haplotype with sequences of *C. galioides* DC. and *C. aspera* Willd. Three samples of *C. gymnophylla* represented a single haplotype that differed from the nearest (*C. gymnophylla* MN793052) by one substitution.

## 4. Discussion

Eight species of charophytes from two genera, *Chara* and *Tolypella,* are reliably known from Dagestan. Seven species are reported here for the first time in the area studied. The investigation of the biodiversity of charophytes of Dagestan using a polyphasic approach allowed us to conduct precise taxa identification of *Chara* species. Morphological traits of oospores studied with SEM were considered. They are in good agreement with species variability [40,52,77,96]. The images of gyrogonites and the surface of the oospore of *C. neglecta* were taken for the first time.

The overall topology of our *rbc*L tree (Figure 6) was similar to that presented in the previous studies of the genus *Chara* [37,38,41,97,98]. We somewhat extended taxon sampling in the genus by adding sequences for *C. neglecta*. Neither cryptic nor new species were found in Dagestan, and all *Chara* accessions were resolved within already-known haplotypes.

The subsection *Hartmania* R.D. Wood of the genus *Chara* is widely known as "*crux et scandalum botanicorum*." In other words, it represents a group of species with still debatable and uncertain boundaries [52,77]. The molecular markers applied so far failed to be helpful in this case [27,31,97,98,100] except for *C. globata*, a species delineable with *rbc*L sequences [26,35], in this study.

Some further perspectives could be outlined based on the phylogenetic data presented here. It was found that *C. neglecta* is undistinguishable from *C. galioides* and *C. aspera* despite well-pronounced differences between these species in stem cortex arrangement (e.g., weakly tylacanthous to isostichous triplostichous in *C. galioides* and weakly aulacanthous diplo-triplostichous to isostichous triplostichous in *C. neglecta* [47,77]). A detailed study on *C. neglecta* phenotypic plasticity is required and will be conducted by the authors in the future with more populations for evaluation of its delineation or merging with *C. galioides*.

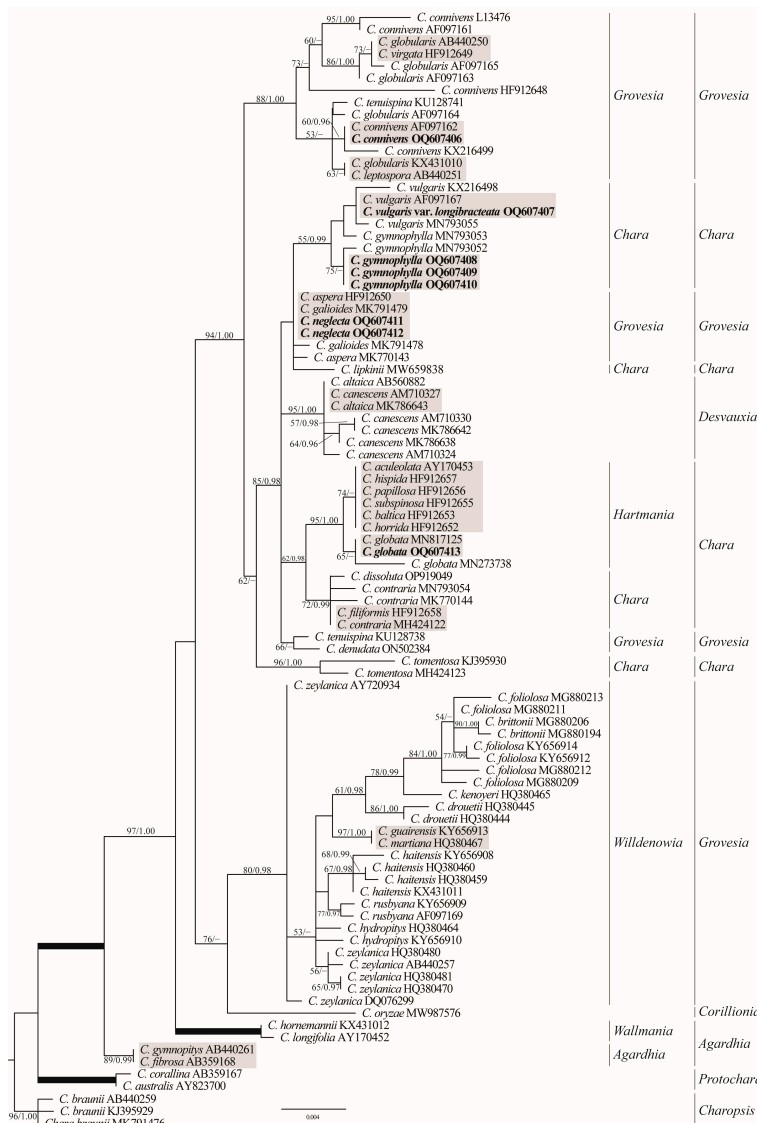

**Figure 6.** Maximum likelihood phylogenetic tree inferred in PAUP with the GTR+I+G nucleotide substitution model from 92 *rbc*L sequences of *Chara*. ML BP (>50%) and BI PP (>0.95). Branches received 100% BP and 1.00 PP support, and the newly obtained sequences are shown in bold. Sequences carrying one genotype are marked with grey. *Chara* sections and subsections are based on [99] with changes from [15,38].

There are 25 species of charophytes known in the Caucasus (Table 1), but some species records require confirmation. Georgia has the richest charophyte flora among the compared regions. The species lists of Caucasian regions largely overlap but are not identical to each other. Seven species known from Dagestan represent less than a third of the species richness of the Caucasus. This number seems low, and new species records are expected according to the species distribution in the Caucasus and neighboring regions (Table 1). The species composition of charophytes from Dagestan is similar to those from Caucasian regions, whereas Azerbaijan and Georgia, as well as Armenia and Dagestan, are closer to each other (Figure 7).

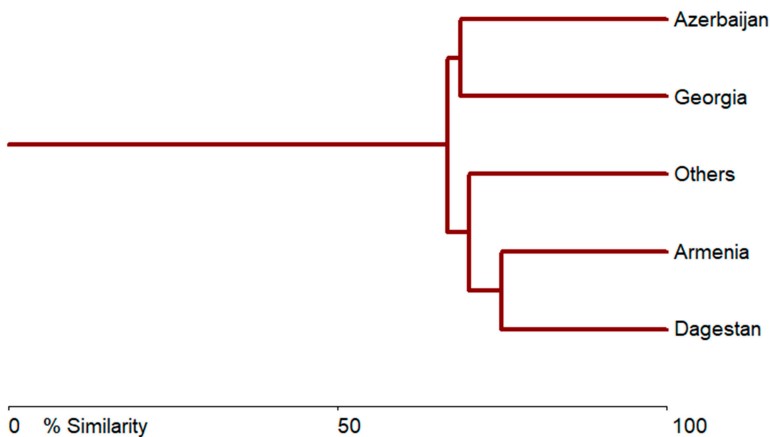

**Figure 7.** The similarity of charophytes species composition of different regions of the Caucasus according to Bray–Curtis cluster analysis (single link) based on reliable records; others: Russian regions of the Caucasus excl. The Republic of Dagestan: Chechen Republic, Kabardino–Balkarian Republic, Karachayevo–Circassian Republic, Krasnodar Territory, Republic of Adygeya, Republic of Ingushetia, Republic of North Ossetia–Alania, Stavropol Territory.

**Table 1.** Species of charophytes from different regions of the Caucasus.

| Species | Dagestan [1] | Others [2] | Georgia [3] | Armenia [4] | Azerbaijan [5] |
|---|---|---|---|---|---|
| *Chara baltica* (Hartm.) Bruzelius | – | + | – | – | (+) |
| *C. braunii* C.C.Gmel. | – | – | + | – | + |
| *C. canescens* Desv. and Loisel. in Loisel. | – | + | + | + | + |
| *C. connivens* Salzm. ex A.Braun | + | + | + | + | + |
| *C. contraria* A.Braun ex Kütz. | + | + | + | + | + |
| *C. denudata* A.Braun | – | – | – | – | + |
| *C. globata* Migula | + | + | + | + | – |
| *C. globularis* Thuill. | + | + | + | + | + |
| *C. gymnophylla* A.Braun | + | + | + | + | + |
| *C. hispida* L. | – | – | + | – | (+) |
| *C. neglecta* Hollerbach | + | – | – | – | + |
| *C. papillosa* Kütz. | – | – | + | (+) | + |
| *C. squamosa* Desf. | – | + | – | – | – |
| *C. strigosa* A.Braun | – | – | – | – | (+) |
| *C. tomentosa* L. | – | (+) | – | – | – |
| *C. vulgaris* L. | + | + | + | + | + |
| *Lamprothamnium papulosum* (Wallr.) J.Groves | – | + | + | – | + |
| *Nitella capillaris* Krock. | – | – | + | – | + |
| *N. flexilis* (L.) C.Agardh | – | + | – | – | – |
| *N. mucronata* (A.Braun) Miq. | – | + | + | – | – |
| *N. opaca* (C.Agardh ex Bruzelius) C.Agardh | – | – | + | – | – |
| *Nitellopsis obtusa* (Desvaux) J.Groves | – | – | + | – | – |
| *Sphaerochara prolifera* (Ziz ex A.Braun) Soulié-Märsche | – | – | + | – | – |
| *Tolypella glomerata* (Desv. in Loisel.) Leonh. | – | – | + | + | + |
| *T. nidifica* (O.F.Müll.) A.Braun | + | + | + | – | – |
| Number of species | 8 | 13(14) | 18 | 6(7) | 13(16) |

[1] A comprehensive bibliography is impossible to report within the scope of this article, so the cited references are those that include all known species from the territory. [1] this study, [2] Russian regions of the Caucasus excl. the Republic of Dagestan: Chechen Republic, Kabardino–Balkarian Republic, Karachayevo–Circassian Republic, Krasnodar Territory, Republic of Adygeya, Republic of Ingushetia, Republic of North Ossetia–Alania, Stavropol Territory [63,80,81,99,101–106], [3] [80,106–109], [4] [77,80,81,107,110–113], [5] [41,79–81,101,107,114–117]. (+) –: Other Russian regions of the Caucasus–records need confirmation with specimen study; existence of vouchers is unknown; Armenia–a record of *C. papillosa* from Lake Sevan needs confirmation because it could actually be based on misidentification of *C. globata*; a record of *C. connivens* from Lake Sevan needs checking because of similarity of its images to *C. aspera* [cf. 81]; Azerbaijan–the recent record of *C. baltica* [81,117] needs to be confirmed with an independent study of the specimen; the old record of *C. hispida* [7] needs confirmation because it could be based on another species.

A few old species records [80] and specimens could not be georeferenced, even up to administrative regions or the whole territory of the Caucasus, but they could be evidence of the possible occurrence of some species in the studied area. The collection by F.A. Marschall von Bieberstein (1768–1826) stored in LE (LE 01157109–01157113) and checked by the first author contained *Chara* cf. *globularis* Thuill., *C. vulgaris*, *Nitella opaca* (C.Agardh ex Bruzelius) C.Agardh and *Tolypella glomerata* (Desv. in Loisel.) Leonh. Some of them were collected in "*deserto Cumanum*" [118], i.e., in an arid region of the drainage basin of the Kuma River that belongs to the Dagestan, Kalmykia, and Stavropol Territory of Russia. The exact georeferencing of these specimens is not possible. Old records of *C. aspera* and *C. tomentosa* L. from the Lake Atu–Kol of the Terek Oblast of the Russian Empire [107] could belong to the current territory of the Republic of Kalmykia. The occurrence of some species of *Nitella* C.Agardh, *Sphaerochara* Mädler, *Nitellopsis obtusa* (Desvaux) J.Groves and *Lamprothamnium papulosum* (Wallr.) J.Groves seems to be possible in Dagestan, although they are rarely found in West Asia [101,119,120].

Despite a long interval of collecting, 1861–2020, only 22 localities are still known from this region, which could be explained by the fact that the charophytes were a neglected group in this region for a long time. The new species records from Dagestan improve species distribution ranges, filling the gaps and changing their outlines in several cases. New localities of *C. connivens* are situated between a few localities in the South Caucasus and the Lower Volga region [45,77,78,101,107].

New localities of *C. contraria, C. globularis,C. vulgaris*, and, especially, *C. gymnophylla* add essential details to their distribution in the Caucasus. *Chara contraria, C. globularis*, and *C. vulgaris* are widely distributed species. They are generalists in many temperate regions [121]. *Chara gymnophylla* is the second most frequently found species in the area studied. It has a wide distribution area in Eurasia, but most records are concentrated in the Mediterranean region and the Middle East [49,77,101,122,123]. It is the second most common species in Iran [49] and northern Israel [123]. Its records are unknown north of the area studied [77,101], i.e., in the southeastern margin of Eastern Europe, and its northern distribution range can be tentatively outlined with the Piedmont area of the North Caucasus.

*Chara globata* is mainly a Central Asian species with several localities in Southeast Europe, North Africa, the Middle East, and China [26,35,49,75,101,124]. Its distribution area extends north of the Caucasus to the arid regions of Volga and Don Interfluve [75,107]. It is known from a few localities in the Caucasus, although its remarkable habit and size nearly exclude its undersampling during targeted searches for charophytes and allow it to be used as a flagship species for biogeographical studies [26]. The records from small water bodies on the coast of the Caspian Sea add a new type of habitat known for this species, mostly associated with much bigger permanent lakes [77,124].

*Chara neglecta* is a species mostly known from the south of Eastern Europe and Central Asia [77]. In the Caucasus, it was formerly known only in Azerbaijan [96,115]. New records in Dagestan point towards its wider distribution in coastal regions of the Caspian Sea.

*Tolypella nidifica* is a species occurring mostly in coastal regions of Europe, where it is mostly known from North and West Europe as well as the West Mediterranean [77]. Only two reliable small areas are known from the North Black Sea region [77,101,104,125,126], and one recent record from inland Georgia by V.S. Vishnyakov [81,112], https://www.inaturalist.org/observations/156074494, accessed on 13 July 2023. The record from the inland of Crimea [53,126] belongs to *Sphaerochara prolifera*, according to published images. Lake Issyk Kul is the sole locality for this species in Central Asia [96,101]. The unexpected new record from Dagestan is the first for the Caspian Sea region and fills the gap in the eastern part of the species distribution area in Eurasia.

The combination of species with distribution areas mainly in the center of Eurasia, such as *C. globata* and *C. neglecta* with mainly Mediterranean–Middle East *C. gymnophylla*, is a notable trait of charophyte flora of the area studied. It seems to be a trait common to other Caucasian regions.

The habitat preference of species recorded from the area studied allows tentative suggestion of two main groups of habitats having dissimilar species compositions. *Chara contraria*, *C. gymnophylla*, and *C. vulgaris* have been found mostly in freshwater small water bodies usually associated with rivers. *Chara globata*, *C. neglecta*, and *Tolypella nidifica* are known only from small brackish water bodies at the coast of the Caspian Sea. They grew together here with *Ruppia maritima* L. during our survey, which evidently indicates a brackish environment. *Chara connivens* and *C. globularis* fall outside this scheme because they are known only from two large artificial mountain and coastal water reservoirs. Surveys of the region, especially of the oligotrophic mountain, brackish coastal, and lowland temporal spring water bodies, seem to be the most fruitful in further clarification of the biogeography of charophytes in the North Caucasus.

Based on habitat preference and distribution in the Caucasus, recommendations can be given to protect some of the species. Coastal water bodies threatened from both land and sea sides are some of the most endangered and shrinking habitats worldwide [127,128]. There is no assessment of charophyte habitat loss at the coast of the Caspian Sea available, but a long-term negative trend is notable for the Black Sea region, where it has already resulted in a significant decrease in charophyte biomass and their community areas [47,129–131]. Some lagoons in this region harbor gyrogonites and oospores in bottom sediments, whereas charophyte stands seem to be lacking [132,133]. The evident loss of coastal habitat can be recognized as too late for the restoration of the initial state [134] and may result in the irreversible disappearance of brackish water charophytes [135]. Charophyte communities of Albufera de València lagoon in the West Mediterranean vanished due to long-lasting and still ongoing eutrophication, although viable oospores and gyrogonites were stored in bottom sediments, leaving a small hope for future restoration of aquatic vegetation [136]. The hydrological change, habitat destruction, and expansion of reed stands because of a decline in livestock grazing can lead to similar results in brackish environments [137,138]. Therefore, *Chara globata*, *C. neglecta*, *Tolypella nidifica*, and their habitats are suggested here as primary targets for charophyte protection in Dagestan. The same need to protect coastal brackish habitats of charophytes was highlighted for Scotland [137], Great Britain [139], the Baltic Sea [138], France [140], and Sardinia [10].

The territory of Dagestan was recognized as a floristic province at the beginning of the study of its flora. It is one of the key areas for the speciation of xerophytic flora in the Caucasus, which can be proven by the high number of local endemics of magnoliophytes [141]. The absence of endemic species among charophytes from Dagestan and the Caucasus disagrees with the flora of terrestrial magnoliophytes, which have a large number of endemic species [142–144]. In contrast, the flora of the submersed aquatic magnoliophytes of this area consists of mostly widely distributed species [111,143–146], although recent speciation of hygrophyte species of *Cardamine* L. (Brassicaceae) probably driven by both geographic separation and ecological divergence was revealed [147]. Therefore, the aquatic ecosystems of this region do not seem to be an area with ongoing diversification and speciation of submersed aquatic macroscopic plants. This paradox looks similar to Tajikistan, which has no unique species of charophytes in contrast with its original flora of terrestrial magnoliophytes [55].

**Supplementary Materials:** The following supporting information can be downloaded at https://www.mdpi.com/article/10.3390/environments10090153/s1, Supplementary Materials: Species name, GenBank accession number, and the haplotypes for the taxa used in our analyses. The sequences obtained in this study are in bold. Shared haplotypes are highlighted in yellow.

**Author Contributions:** Conceptualization, R.E.R.; methodology, R.E.R., M.M.M. and S.B.; software, R.E.R., S.B., V.Y.N. and A.A.G.; validation R.E.R., M.M.M., S.B., V.Y.N. and A.A.G.; formal analysis, R.E.R., M.M.M., S.B., V.Y.N. and A.A.G.; investigation, R.E.R., M.M.M., S.B., V.Y.N. and A.A.G.; data curation, R.E.R., V.Y.N. and A.A.G.; writing—original draft preparation, R.E.R., M.M.M., S.B., V.Y.N. and A.A.G.; writing—review and editing, R.E.R., M.M.M., S.B., V.Y.N. and A.A.G.; visualization, R.E.R., M.M.M., S.B., V.Y.N. and A.A.G.; funding acquisition, R.E.R., S.B., V.Y.N. and A.A.G. All authors have read and agreed to the published version of the manuscript.

**Funding:** This research was funded by the project "Flora and taxonomy of algae, lichens and bryophytes in Russia and phytogeographically important regions of the world" (no. 121021600184-6) of the Komarov Botanical Institute of the Russian Academy of Sciences, within the state assignment of Ministry of Science and Higher Education of the Russian Federation (theme No. 121031000117-9), with the financial support the Ministry of Science and Higher Education of the Russian Federation under the agreement dated 28 September 2021, no. 075-15-2021-1056 (placement of the algal specimens in the LE collection), and the Israel Ministry of Aliyah and Integration.

**Data Availability Statement:** The data presented in this study are available on request from the corresponding author. In addition, data that support the findings of this study are openly available in GenBank.

**Acknowledgments:** L.A. Kartseva is kindly acknowledged for her assistance in SEM studies, I.V. Tatanov for his encouragement and guidance with historical herbarium collection, V.V. Vishnyakov and S.V. Smirnova for inaccessible reference copies, anonymous reviewers for their encouragement and kind suggestions for the improvement of the manuscript.

**Conflicts of Interest:** The authors declare no conflict of interest.

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
