# Peer review of "The Charophytes (Charophyceae, Characeae) from Dagestan Aquatic Habitats, North Caucasus: Biogeographical and Barcoding Perspectives"

_environments, doi:10.3390/environments10090153_

Round 1

Reviewer 1 Report

This publication adds valuable knowledge about occurrence of charophytes in a region that up to now has been poorly investigated. Data collection and analysis was done carefully, methods are adequate, the results are well presented, the discussion is most reasonable and interesting also in a global context, especially the failure to find endemic submerged plants in a region that is characterized by a high share of endemcs among terrestrial plants. The language is understandable, but a thorough language check would considerably improve the quality of this paper: Some few suggestions are given below, but this list is far from complete!

Line 18: insert and“ behind “Caucasus“

line 58: replace compositions“ by composition“

line 116: replace Charophytes“ by Charophyte“

line 196: replace slow“ by slowly“.

Line 375: remove in“

line 380 – what is meant – really temporal“, not temperate“?

Line 438f: I suggest to replace The suggestion of some species for protection is possible based on the habitat preference and distribution in the Caucasus“ by Based on habitat preference and distribution in the Caucasus, recommendations can be given to protect some of the species“ (or something similar): see also Abstract, line 30f!

fig. 1B: symbol 6 is covered by another symbol and therefore, hardly visible – please improve!

Taxonomy is said to follow the most recent literature (see line 75f): In Fig. 6, Chara intermedia, C. polyacantha and C. rudis should therefore be named C. papillosa, C. aculeolata and C. subspinosa, respectively. Please check also other taxa names for nomenclature!

Line 299f: Failure to separate Hartmania taxa genetically from each other was also described by: Nowak, P. & H. Schubert, 2019. Genetic variability of charophyte algae in the Baltic Sea area. Botanica Marina 62: 75-82. This reference could (!) be added.

I recommend to remove Fig. 7b: Apart from the fact that this Fig. does not give any relevant information not already shown in Fig. 7a, it does not make sense to present records that are not reliable.

A need to protect brackish habitats along the coast is also mentioned by: Becker, R., 2019. The Characeae (Charales, Charophyceae) of Sardinia (Italy): habitats, distribution and conservation. Webbia 74: 83-101. Also this reference could be added.

see above

Author Response

Thank you very much for your kind suggestions for the improvement of the manuscript!

Rev 1 Report 1

Open Review

Quality of English Language

( ) I am not qualified to assess the quality of English in this paper
( ) English very difficult to understand/incomprehensible
( ) Extensive editing of English language required
(x) Moderate editing of English language required
( ) Minor editing of English language required
( ) English language fine. No issues detected

Yes

Can be improved

Must be improved

Not applicable

Does the introduction provide sufficient background and include all relevant references?

(x)

( )

( )

( )

Are all the cited references relevant to the research?

(x)

( )

( )

( )

Is the research design appropriate?

(x)

( )

( )

( )

Are the methods adequately described?

(x)

( )

( )

( )

Are the results clearly presented?

(x)

( )

( )

( )

Are the conclusions supported by the results?

(x)

( )

( )

( )

Comments and Suggestions for Authors

This publication adds valuable knowledge about occurrence of charophytes in a region that up to now has been poorly investigated. Data collection and analysis was done carefully, methods are adequate, the results are well presented, the discussion is most reasonable and interesting also in a global context, especially the failure to find endemic submerged plants in a region that is characterized by a high share of endemcs among terrestrial plants. The language is understandable, but a thorough language check would considerably improve the quality of this paper: Some few suggestions are given below, but this list is far from complete! – English spelling was checked by native speaker (see the certificate attached)

Line 18: insert “and“ behind “Caucasus“ - changed

line 58: replace “compositions“ by “composition“ - changed

line 116: replace “Charophytes“ by “Charophyte“ - changed

line 196: replace “slow“ by “slowly“. - changed

Line 375: remove “in“ - changed

line 380 – what is meant – really “temporal“, not “temperate“? - changed

Line 438f: I suggest to replace “The suggestion of some species for protection is possible based on the habitat preference and distribution in the Caucasus“ by “Based on habitat preference and distribution in the Caucasus, recommendations can be given to protect some of the species“ (or something similar): see also Abstract, line 30f! - changed

fig. 1B: symbol 6 is covered by another symbol and therefore, hardly visible – please improve! - changed

Taxonomy is said to follow the most recent literature (see line 75f): In Fig. 6, Chara intermediaC. polyacantha and C. rudis should therefore be named C. papillosaC. aculeolata and C. subspinosa, respectively. Please check also other taxa names for nomenclature! - improved

Line 299f: Failure to separate Hartmania taxa genetically from each other was also described by: Nowak, P. & H. Schubert, 2019. Genetic variability of charophyte algae in the Baltic Sea area. Botanica Marina 62: 75-82. This reference could (!) be added. - added

I recommend to remove Fig. 7b: Apart from the fact that this Fig. does not give any relevant information not already shown in Fig. 7a, it does not make sense to present records that are not reliable. - removed

A need to protect brackish habitats along the coast is also mentioned by: Becker, R., 2019. The Characeae (Charales, Charophyceae) of Sardinia (Italy): habitats, distribution and conservation. Webbia 74: 83-101. Also this reference could be added. - added

Comments on the Quality of English Language

see above

Reviewer 2 Report

The article deals with an important topic and provides information on charophytes in one relatively large region of Caucasus. Without going into the strengths of the paper, I would like to comment on the weaknesses I have observed.

1. I suggest rewriting the abstract so that it is not in the style of an advertisement, but introduces the reader to the problem, the main points of the methodology, the results and conclusions.

2. In my opinion, the introduction is too short and does not provide an adequate overview of the history of research in the region under consideration, nor of the current trends in charophyte research in Europe, Asia and other regions of the world. The authors do not even mention recent studies and published syntheses.

3. The methodology does not indicate how many herbarium specimens were used for the study. It should be stated.

4. In my opinion, it is completely unnecessary to state each time that a species is new to Dagestan, it would be sufficient to list the species that are new to the region at the beginning of the results.

5. In my opinion, the sub-section of the results on phylogenetic analysis is too short and uninformative. I strongly recommend that not only the dendrogram be presented, but also the significant differences, and that the discussion examine the possible reasons for such differences. 

6. It is not clear why Figure 7 is needed if the same information is provided in Table 1. 

7. I missed the discussion of the results of the phylogenetic analysis. 

8. What are the conservation implications of the results of this study? It would be very important if you would also address this issue in the discussion and compare them in a much wider context. Then the article would really fit the theme of the journal. The article needs to be restructured in order to put the regional study in a wider context. 

Should be improved

Author Response

Thank you very much for your kind suggestions for the improvement of the manuscript!

Rev 2 Report 1

Open Review

Quality of English Language

( ) I am not qualified to assess the quality of English in this paper
( ) English very difficult to understand/incomprehensible
( ) Extensive editing of English language required
(x) Moderate editing of English language required
( ) Minor editing of English language required
( ) English language fine. No issues detected

Yes

Can be improved

Must be improved

Not applicable

Does the introduction provide sufficient background and include all relevant references?

( )

( )

(x)

( )

Are all the cited references relevant to the research?

( )

( )

(x)

( )

Is the research design appropriate?

( )

(x)

( )

( )

Are the methods adequately described?

( )

( )

(x)

( )

Are the results clearly presented?

( )

(x)

( )

( )

Are the conclusions supported by the results?

( )

( )

(x)

( )

Comments and Suggestions for Authors

The article deals with an important topic and provides information on charophytes in one relatively large region of Caucasus. Without going into the strengths of the paper, I would like to comment on the weaknesses I have observed.

  1. I suggest rewriting the abstract so that it is not in the style of an advertisement, but introduces the reader to the problem, the main points of the methodology, the results and conclusions.- rewrited
  2. In my opinion, the introduction is too short and does not provide an adequate overview of the history of research in the region under consideration, nor of the current trends in charophyte research in Europe, Asia and other regions of the world. The authors do not even mention recent studies and published syntheses. - improved
  3. The methodology does not indicate how many herbarium specimens were used for the study. It should be stated. - added
  4. In my opinion, it is completely unnecessary to state each time that a species is new to Dagestan, it would be sufficient to list the species that are new to the region at the beginning of the results. - changed
  5. In my opinion, the sub-section of the results on phylogenetic analysis is too short and uninformative. I strongly recommend that not only the dendrogram be presented, but also the significant differences, and that the discussion examine the possible reasons for such differences. - improved
  6. It is not clear why Figure 7 is needed if the same information is provided in Table 1. – Figure 7b was erased. Figure 7 illustrates general similarity/difference between datasets for regions.
  7. I missed the discussion of the results of the phylogenetic analysis. – improved, see beginning of “discussion”
  8. What are the conservation implications of the results of this study? It would be very important if you would also address this issue in the discussion and compare them in a much wider context. Then the article would really fit the theme of the journal. The article needs to be restructured in order to put the regional study in a wider context. - improved

Reviewer 3 Report

It was with real pleasure that I studied the manuscript describing the results of research on the charophyte flora in the republic of Dagestan, which was posted in the form of a preprint on July 19, 2023 on the multidisciplinary platform Preprints.org. Detailed research included a query of scientific literature, a review of the herbarium resources at the Komarov Botanical Institute of RAS along with the verification of the markings of the specimens collected there, and field research. The manuscript describes all sites of charophyte species, along with their location, and the list of species is supplemented with very good photographic documentation of specimens and SEM photos of several oospores. The determination was made on the basis of morphological features and genetic tests - the sequence of the rbcL gene, which allowed to verify the determination. The conducted research can be considered as a model, which in my opinion fully authorizes the recommendation of the manuscript for publication.

Here are some remarks:

Line 43: I propose “The DNA barcoding .....”

Line 124: The ecoregions are marked in Figures 1b and 1c.

Line 140: What does “4000’” mean?

Line 180: I suggest giving the approximate elevation in meters above sea level in parentheses.

Line 186: Point “3.” was already in line 182.

Line 218: I propose “(Under the name C. foetida f. condensata A.Braun; Vilhelm, 1930).”

Line 229: As in line 218.

Round 2

Reviewer 2 Report

The manuscript has improved very significantly since the first peer review and the authors' revisions. 

1. It is usual in English to give centuries in Arabic rather than Roman numerals (line 87). 

2. As the herbarium label entries are translated and adapted, I would recommend that the dates of collection of specimens should also be given according to the English standard (e.g. 2 Aug 1888), or in another clear format that is acceptable to the authors. In any case, the format of the dates should be standardised (see line 254). 

Minor editing required

Author Response

Dear Reviewer 2,

Thank you for your comments.

Please find our responses to each comment below.

With best regards,

Sophia Barinova,

Comments and Suggestions for Authors

The manuscript has improved very significantly since the first peer review and the authors' revisions. 

  1. It is usual in English to give centuries in Arabic rather than Roman numerals (line 87). 

Response: corrected

  1. As the herbarium label entries are translated and adapted, I would recommend that the dates of collection of specimens should also be given according to the English standard (e.g. 2 Aug 1888), or in another clear format that is acceptable to the authors. In any case, the format of the dates should be standardised (see line 254). 

Response: corrected

 Comments on the Quality of English Language

Minor editing required

Response: corrected

Round 3

Reviewer 2 Report

No additional comments on the quality of manuscript.

Minor editorial revisions